# Towards a Grounded Theory of Causation for Embodied AI

**Taco Cohen**[1]

[1]Qualcomm AI Research[*]

## Abstract

There exist well-developed frameworks for causal modelling, but these require rather a lot of human domain expertise to define causal variables and perform interventions. In order to enable autonomous agents to learn abstract causal models through interactive experience, the existing theoretical foundations need to be extended and clarified. Existing frameworks give no guidance regarding variable choice / representation, and more importantly, give no indication as to which behaviour policies or physical transformations of state space shall count as interventions. The framework sketched in this paper describes actions as *transformations* of state space, for instance induced by an agent running a policy. This makes it possible to describe in a uniform way both transformations of the micro-state space and abstract models thereof, and say when the latter is veridical / grounded / natural. We then introduce (causal) variables, define a *mechanism* as an invariant predictor, and say when an action can be viewed as a "surgical intervention", thus bringing the objective of causal representation & intervention skill learning into clearer focus.

## 1 INTRODUCTION

Most work in causal inference is aimed at helping scientists make causal judgements, particularly when this is difficult due to lack of interventional data and confounding [Pearl, 2009, Peters et al., 2017]. In such applications, there is usually a fairly clear idea about the meaning of the causal variables (e.g. employment rate, cholesterol level, etc.), and some intuitive understanding of what is meant by "intervention" (e.g. raise minimum wage, provide treatment, etc.).

As is clear from these examples, one typically relies on a lot of human perception capabilities, concepts, knowledge, and skills, which are not available to an autonomous agent learning about its environment through interaction. Such an agent must learn not only a causal representation [Schölkopf et al., 2021], but as we argue here, also a set of *intervention skills* (policies/options [Sutton et al., 1999]) to set mechanisms for causal variables, whenever possible. Furthermore, it would be nice if these intervention skills were "surgical", so that they enable simple SCM-like causal reasoning.

Here one runs into foundational issues that must be cleared up before we can get to work. When can a policy be seen as an intervention that sets a mechanism for a variable, what does it mean to do so "surgically", and what does it even mean to say that a mapping is a "mechanism"? A common view is that intervention in MDPs means that the agent chooses a low-level action at each time step, but this does not lead to very interesting or meaningful interventions in case the actions are e.g. a robot's motor commands.

We propose (Sec. 2) to model actions as transformations $\text{do}_X(a) : X \to X$ of a state space $X$ induced by running a policy $a$, and consider a process $\text{proc}_Y : X \to Y$ that happens after taking some actions. The agent also has a model of the actions and process, which can be seen as an abstraction (i.e. natural transformation) of the underlying system dynamics (Sec. 2.2). Like SCMs, our models are essentially deterministic but one can easily incorporate uncertainty by putting distributions on noise variables and pushing them forward through deterministic maps (Sec 2.3).

In Sec. 3 we introduce (causal) variables $Y_i$, and show when the actions $\text{do}_X(a)$ behave as surgical interventions that set a mechanism for a variable. We show how one can encode any SCM in our framework, so nothing is lost and our framework really does capture "causality". The ideas are illustrated using the example of a robot with arm and camera, manipulating a set of dominoes. In Sec. 4 we discuss related work along with implications for causal representation & intervention skill learning and foundations of causality.

---

[*]Qualcomm AI Research is an initiative of Qualcomm Technologies, Inc.

*Accepted for the Causal Representation Learning workshop at the 38th Conference on Uncertainty in Artificial Intelligence* (UAI CRL 2022).

## 2 MODELS OF ACTIONS & OUTCOMES

Consider an agent interacting with an environment. Assuming that any stochasticity arises from partial observability, we can model both agent and environment as deterministic functions env : $E \times A \to E \times O$ and agent : $O \times M \to A \times M$ (policy), where $E$ is the environment state, $A$ the action, $O$ the observation, and $M$ the agent memory state. Composing these functions appropriately, we obtain a map $X \to X$ where $X = E \times A \times M$, as shown below. This map tells us what happens to $X$ when we run the policy defined by agent for one step. More generally, running the policy $a$ for a fixed number of steps, or until some termination condition is met (as in the options framework[1] Sutton et al. [1999], Precup [2000]), we obtain a map $\mathrm{do}_X(a) : X \to X$.

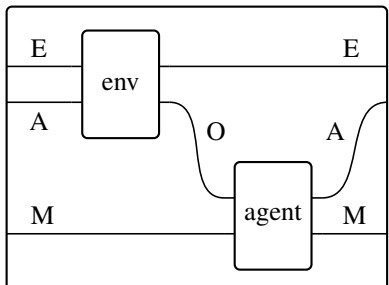

From hereon, we will abstract away from the details of the agent/environment loop and simply discuss policies/actions/options/skills/interventions $a$ and the mappings $\mathrm{do}_X(a) : X \to X$ that they induce. We will assume that our agent has some elementary actions $a, b, \ldots$ called generators. We can do one action after another, so we also have composite actions $a = bc$ (first $c$, then $b$), and the corresponding mapping is defined by function composition: $\mathrm{do}_X(bc) = \mathrm{do}_X(b)\,\mathrm{do}_X(c)$ (Throughout this paper, we will omit the composition symbol $\circ$). For the characteristic phenomena of causality to arise, it is necessary to consider a process that happens after acting, which is also modeled as a map $\mathrm{proc}_Y : X \to Y$. This map produces for each state in $X$ an *outcome* in $Y$. Note that at this point $X$ and $Y$ are bare sets, not to be thought of as consisting of variables.

We may or may not be able to choose to initiate the process, but in any case we assume that during the process we have no ability or intention to intervene or observe. We would still like to influence the outcome though (because the outcome may have a *value* to us), which we could do by first performing actions that "change the mechanisms" determining the outcome. Examples include giving a patient treatment (and then letting the physiological process unfold), removing one domino from a chain before initiating the process by pushing another, etc. We would like to emphasize that one may very well wish to reason about taking actions during or after $\mathrm{proc}_Y$, but here our only objective is to find a minimalistic setup where we can study grounded causal reasoning.

---

[1] Unlike general options, we assume for simplicity that all actions can always be performed

The outcome of taking an action is what we get by doing the action and then running the process:

**Definition 2.1** (Outcome). *The outcome of action $a$ is:*

$$\mathrm{outcome}_Y^a = \mathrm{proc}_Y\,\mathrm{do}_X(a) : X \to Y. \tag{1}$$

Note that the outcome is a *map*, because the answer to the question "what outcome do I get if I $\mathrm{do}(a)$?" depends on the state. Hence we trivially have counterfactual / rung 3 content built in [Pearl and Mackenzie, 2018, Bareinboim et al., 2022]; a map tells us what the output *would be* for any input. Outcome maps are similar to potential response maps $Y_a(u)$ of SCMs (Pearl [2009], Ch. 7), which depend on exogenous variables $u$ and intervention $a$ (which determines the active mechanisms). However, as we explain shortly, in our setup the state determines both the exogenous variables and the mechanisms that are active, and interventions act on the latter. Indeed it is clear that whatever is meant by "mechanism", it must be something that depends on the state. This is because the "mechanisms" are supposed to determine the outcome of the process, and both classical physics and our agent/environment model tell us that actually the future outcome is determined by the present state.

Let us summarize the discussion above with a definition:

**Definition 2.2** (Action Model). *An action model* $\mathcal{M} = (X, Y, A, \mathrm{proc}_Y)$ *consists of a set of states* $X$, *outcomes* $Y$, *a process* $\mathrm{proc}_Y : X \to Y$ *and a collection* $A$ *of generator actions* $\mathrm{do}_X(a) : X \to X$ *(including the identity* $\mathrm{do}_X(id) = id_X$*), and all composites of these maps. Composite actions are denoted* $\mathrm{do}_X(ab) = \mathrm{do}_X(a)\mathrm{do}_X(b)$, *and outcomes by* $\mathrm{outcome}_Y^{a_1 \cdots a_n} = \mathrm{proc}_Y\mathrm{do}_X(a_1 \cdots a_n)$.

### 2.1 RUNNING EXAMPLE: ROBO-DOMINOES

One example that we will use throughout the paper is that of a robot arm manipulating a configuration of dominoes using visual input from a camera and skills/policies that execute low-level motor actions. Here $X$ is some set of physical configurations of dominoes (plus agent state) where the model is deemed applicable, and the actions/skills could include: setting up the dominoes in a particular configuration (initialization, e.g. to a chain, tree, or loop), putting a barrier between dominoes / removing one, picking / placing a domino, moving a domino to some position, and designating (e.g. in agent memory) a domino to be pushed and the direction of pushing ("choosing a push"). The process $\mathrm{proc}_Y$ consists of pushing the designated domino and waiting for everything to fall and then recording the state. Note that here $Y$ is a subset of $X$, but none of our results will depend on this.

It is intuitively obvious that these actions, when executed skillfully, are surgical interventions. Much of the rest of this paper is dedicated to elucidating the general mathematical properties satisfied by such actions that justify this

interpretation. One may already notice that interventions on (what we intuitively think of as) independent mechanisms commute, while interventions targeting the same variable overwrite. Finally, note that although one can think about a particular domino setup (with chosen push) as a causal graph, one cannot easily describe even this relatively simple domain using a single graph. For instance, when we designate a different domino and direction to be pushed, many arrows can change / reverse, completely changing the graph. Similarly, judiciously turning a domino at a fork that leads to a loop can reverse all the arrows in the loop. So we see that although graphs can play an important role in causal reasoning about actions, common-sense reasoning capabilities require a more general kind of structure (e.g. a small category). Before we begin our discussion of causal variables and mechanisms though, we first need to discuss the relation between our actions as policies and the agent's model thereof.

## 2.2 NATURAL MAPS BETWEEN MODELS

Usually we do not have access to the full state and outcome, and we are not interested in modelling the system in complete detail. For instance, dominoes can be described in an arbitrarily detailed manner, but normally we only care about if and which way they fall. Let us therefore denote the unknown true system $\overline{\mathcal{M}} = (\overline{X}, \overline{Y}, \overline{A}, \mathrm{proc}_{\overline{Y}})$ whose actions are induced by agent policies, and a simplified model $\mathcal{M} = (X, Y, A, \mathrm{proc}_Y)$ that the agent can use to reason about the outcome of actions. One can think of $\overline{X}$ as the computer memory state of a simulation, or a classical physical state, and $X, Y$ as latent/representation spaces.

The specification of these sets and maps completely defines the two models, but theoretically the job of modelling is not done until we specify how the model state and outcome ought to be related to the system state and outcome. For this we introduce maps $x : \overline{X} \to X$ and $y : \overline{Y} \to Y$. For $\mathcal{M}$ to be a perfectly accurate abstraction of $\overline{\mathcal{M}}$ (i.e. to be veridical), $x$ and $y$ should define a natural transformation between the two models, which means that $\forall a \in A$ the following diagram commutes:

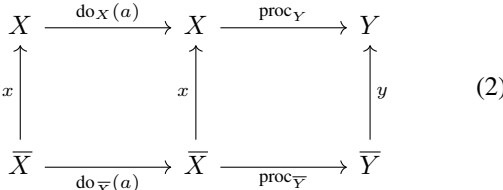

$$(2)$$

That this diagram commutes means that if we follow two directed paths from one node to another, the corresponding composite maps are equal. For instance, we can see that $x \, \mathrm{do}_{\overline{X}}(a) = \mathrm{do}_X(a) \, x$ and $y \, \mathrm{proc}_{\overline{Y}} = \mathrm{proc}_Y \, x$. Intuitively the first equation tells us that if we do an action in the system $\overline{X}$ (e.g. by running a policy) and then evaluate $x : \overline{X} \to X$,

we get the same thing as if we first evaluate $x$ and then perform the corresponding action in our model. Similarly, measuring $x$ and then predicting via $\mathrm{proc}_X$ is the same as first running the true process $\mathrm{proc}_{\overline{Y}}$ and then measuring $y$.

We emphasize that in existing causal modeling frameworks, one can only define veridicality in natural language, whereas in our framework it is a mathematical relation between functions. This is because existing frameworks have no analog of the action $\mathrm{do}_{\overline{X}}$ in the micro-state space. Thus, once we define causal variables (Sec. 3), we have precisely defined for the first time what it means for a causal model to be veridical / grounded. The concept of natural transformation can also be used for model abstraction, itself an important topic Abel [2022], Geiger et al. [2021], Chalupka et al. [2016], Beckers et al. [2019], Beckers and Halpern [2019], de Haan et al. [2020]. Our definition automatically captures the idea that the compositional structure of interventions should be preserved Rubenstein et al. [2017]. As is evident from the equation $\mathrm{do}_X(a) \, x = x \, \mathrm{do}_{\overline{X}}(a)$, naturality is a generalization of equivariance, which is the central concept in geometric DL [Cohen, 2021, Bronstein et al., 2021].

## 2.3 UNCERTAINTY AND VIRTUAL ACTIONS

As in classical physics, the model state $X$ contains *all* information necessary to determine $Y$, and both $X$ and $Y$ are deterministic functions of the underlying system micro state/outcome. This is not to say that $X$ or $Y$ are fully observable or that we need to have a perfect predictor $\mathrm{proc}_Y$. In the more likely partially observed scenario, one could endow $\overline{X}$ with the structure of a probability space, and view $x : \overline{X} \to X$ and $y : \overline{Y} \to Y$ as random variables (which are indeed defined as maps in measure-theoretic foundations of probability Rosenthal [2006]). In this paper we will not be concerned with partial observability and beliefs but we note that the correctness of any probabilistic inference about $X$ or $Y$ can only be judged once they are defined as random variables (measurable maps) $x, y$. For instance, even if a domino is visually occluded by some object, it still has a definite physical state that ought to be included in $X$ and $Y$.

Although we have motivated the definition of actions as maps $X \to X$ via policies, one could also admit actions in the model $\mathcal{M}$ for which one does not actually have a policy that implements it in $\overline{\mathcal{M}}$ (indeed, for many maps there will not exist such a policy in a given environment). For instance, people are very well able to consider questions such as "what would happen to the tides if we removed the moon?", without knowing how to actually do the latter [Pearl, 2019]. Which virtual actions are to be admitted is currently not clear, but one might argue only physically possible ones are of interest. It has been suggested that the distinction between possible and impossible transformations is the essential content of physical laws Deutsch [2012], Marletto [2016] (e.g. transformations must conserve energy).

# 3 CAUSAL VARIABLES, MECHANISMS & INTERVENTIONS

Actions change the state and outcome, but intervention is fundamentally about changing *mechanisms*. In order to discuss mechanisms, we need to split the outcome $Y$ into variables, so let us assume that $Y = \prod_{i \in \mathcal{I}} Y_i$ where each $Y_i$ is a set of values for the $i$-th variable (e.g. numbers but not necessarily). The product comes with projection maps $\pi_i : Y \to Y_i$ that forget all variables except $i$, and similarly for sets of variables $I$ we have $\pi_I : Y \to Y_I$ and for subsets $I \subseteq J$ we have $\pi_I^J : Y_J \to Y_I$ satisfying $\pi_I^J \pi_J = \pi_I$. Having defined variables we can consider the outcome of an action on a subset of variables:

$$\text{outcome}_{Y_J}^a \equiv \text{outcome}_J^a \equiv \pi_J \, \text{proc}_Y \, \text{do}_X(a) : X \to Y_J. \tag{3}$$

It is important to note that although the maps $\text{proc}_Y$ and $\text{outcome}_Y^a$ output all the $Y_i$ variables at once, this should not be taken to mean that they describe simultaneous events. We merely assume that at some point in time, when the process has ended, there is a *record* of all the variables Rovelli [2020].

For the unknown and non-factored set of system outcomes $\overline{Y}$, we may assume without loss of generality that all outcomes are possible, i.e. that $\text{proc}_{\overline{Y}} : \overline{X} \to \overline{Y}$ is surjective so that for each outcome there is a state in $\overline{X}$ that results in that outcome (otherwise just restrict $\overline{Y}$). Similarly we shall assume that $x : \overline{X} \to X$ is surjective, i.e. that all model states are possible to obtain. However in general a "disentangled" choice of variables $y : \overline{Y} \to \sqcap_i Y_i$ will not be surjective, so that our model contains "impossible outcomes" – joint assignments to the variables $Y_i$ that can never occur as a result of running $\text{proc}_{\overline{Y}}$ and then evaluating $y$.

For instance, if we want to represent the state of each individual domino by a variable $Y_i$, we will find that each domino can be in every position, but no two dominoes can be in the same position at once, nor is it possible that one domino falls flat while the next one stays upright. Similarly, the ideal gas law says that only certain values of temperature, pressure, and volume are jointly possible for a particular type and amount of gas. So we see that the image of $\text{proc}_Y$, i.e. the set of possible outcomes, represents an important piece of knowledge about our model of the system.

Instead of considering possible outcomes of $\text{proc}_Y$, one can consider the image for any $\text{outcome}_J^a$ map. Notice that $\text{outcome}_Y^a$ is obtained from $\text{proc}_Y$ by precomposition with $\text{do}_X(a)$, and it is a general fact about functions that the act of precomposing a function can only reduce (not increase) the set of possible outcomes (i.e. image). In other words, by taking action before $\text{proc}_Y$, we can make sure that the outcome is in a restricted set of possible outcomes associated with the action.

The observation that disentangled representations often contain impossible joint outcomes has important implications for (causal) representation learning. Indeed, methods such as VAEs with Gaussian priors Kingma and Welling [2013], Rezende et al. [2014] attempt to densely pack the representation space, and learned representations are often evaluated by their ability to interpolate between data points, or recombine different variables from two datapoints (e.g. the hair style from one image and the facial expression from another). Whereas for some intuitively meaningful variables (such as facial expression and hair style, neither of which causes the other, but which can be controlled independently) all combinations are possible (i.e. have independent support; Wang and Jordan [2021]), this is often not the case. So it is clear that in causal representation learning, we should not always aim to fill up the representation space $Y$, nor assume that an ideal representation should allow arbitrary interpolation/recombination operations without venturing into impossible territory.

## 3.1 DETERMINATION & EFFECTIVE ACTIONS

The presence of impossible joint outcomes makes it possible that, even in the absence of subjective probabilities/beliefs, one variable $Y_I$ can have *information* about another variable $Y_J$, in the sense that knowing $Y_I$ rules out certain values for $Y_J$. In general, a subset of a product, e.g. im $\text{outcome}_Y^a \subseteq \prod_i Y_i$, is called a *relation*. When there is for each possible outcome $Y_I$ only one possible outcome $Y_J$, we have a *functional* relation, which we call:

**Definition 3.1** (Determination). *Let $a$ be an action sequence and let $I, J$ be (sets of) variables. We say that $\text{outcome}_J^a$ is determined by $\text{outcome}_I^a$ via $f^a : Y_I \to Y_J$ if the following diagram commutes:*

$$
\begin{array}{ccc}
Y_I & & \\
{\scriptstyle \text{outcome}_I^a}\Big\uparrow & \searrow^{f^a} & \\
X & \xrightarrow[\text{outcome}_J^a]{} & Y_J
\end{array}
\qquad i.e. \text{outcome}_J^a = f^a \, \text{outcome}_I^a
$$

*The determination is unique if there is exactly one such $f^a$, which implies that $\text{outcome}_I^a$ is surjective.*

It is tempting to interpret $a$ as setting mechanism $f^a$ for $Y_J$, but as we will see shortly, determination is necessary but not sufficient for $f^a$ to deserve the name mechanism.

If $\text{outcome}_I^a$ determines $\text{outcome}_J^a$ via $f^a$, then for any $b$:

$$\text{outcome}_J^{ab} = \text{outcome}_J^a \text{do}(b) = f^a \, \text{outcome}_I^{ab}.$$

In other words, the determination relation is invariant to precomposition (doing $b$ *before* $a$). This makes sense because determination says that *wherever we start in $X$*, after $\text{do}_X(a)$ and $\text{proc}_Y$, we can tell the outcome $Y_J$ from $Y_I$ using $f^a$. However, determination relations are in general

not invariant to doing $b$ *after* $a$ (but before $\text{proc}_Y$), and this observation will be key to understanding mechanisms.

In our example, let $Y_i$ be the state of domino $i$ after pushing the designated domino and waiting. An action such as placing domino $i$ at some position will not in general lead to determination, because the outcome for domino $j \neq i$ depends completely on the rest of the state. However, this action might result in determination in some *context*, i.e. a subset $X_c \subseteq X$. For instance, there are sets of states where if newly placed domino $i$ falls, then also domino $j$ falls. The context of being in a state "after $s$" (e.g. initializing) can be described as $X_s = \text{im do}_X(s)$ or simply context $s$.

Initialization itself is an action that satisfies determination relations unconditionally. Whatever configuration was there before, it gets replaced by one of our choice. Perhaps our robot has the skill to set the high-level state to $x$ *exactly*, or maybe the setup will vary a bit based on e.g. actuator noise $Y_I = U$ which we control nor observe, or maybe there is an observable but not controllable instruction $U'$ for how to place the dominoes. In any case, we see that after initialization, usually very many determination relations hold. If the state is exactly $x$, there is one possible outcome $y = \text{proc}_Y x$, and so *every variable determines every other one* in a highly non-unique way. Hence it is clear that determination is necessary but not sufficient to speak of causation and mechanisms.

In SCM theory, one typically considers atomic interventions $\text{do}(V_j = \bar{v}_j)$ that set a variable to a value. A value can be viewed as a map $\bar{v}_j : 1 \to V_j$, where 1 is the one-object set, so we can describe this as a special kind of determination:

**Definition 3.2** (Effectiveness). *We say that $a$ is **effective** at setting $\bar{a}_J : 1 \to V_J$ if $\text{outcome}_J^a$ is determined by the empty product 1 (no variables) via $\bar{a}_J$, i.e. $\text{outcome}_J^a = \bar{a}_J \text{outcome}_0^a$ (where $\text{outcome}_0^a = \pi_0$ is the unique map from $X$ to 1). In simple terms, $\text{outcome}_J^a = \text{const}_{\bar{a}_J}$ is a constant map with value $\bar{a}_J$. We say that $a$ is effective at setting $\bar{a}_J : 1 \to Y_J$ in context $s$, if $\text{outcome}_J^{as} = \text{const}_{\bar{a}_J}$.*

## 3.2 INVARIANT MECHANISMS

The determination relations that we intuitively think of as "mechanisms" hold not just for the outcome maps associated with one intervention $a$ but many. For example, after setting up a domino configuration and choosing a domino to be pushed, it could be that an "ancestor domino" $i$ determines descendant $j$, but a method of predicting outcome $j$ using $i$ does not work anymore if we remove a domino in between the two. However, if $j$ comes right after $i$, then the end state of $j$ is determined by the end state of $i$, and this continues to be true if we perform interventions such as taking away previous or later dominoes, placing barriers (except between $i, j$), or changing the domino to be pushed (though not if we push a downstream domino backwards).

The exceptions we noted here can be thought of as interventions that *change the mechanism(s)*, meaning that the old mechanism becomes obsolete (i.e. cannot be used for prediction anymore), and a new one (with its own invariance properties) is established for the target variable. For an effective (i.e. atomic) intervention, the new mechanism is a constant, while in general it may be any function [Correa and Bareinboim, 2020]. A good (surgical) intervention will thus replace an old mechanism for $Y_i$ by a new one, and furthermore, leave the mechanisms for other variables intact. Furthermore, it would be nice if in the new context (after the intervention), the same invariances hold, so other surgical interventions remain surgical.

Strictly speaking, in our theory "changing the mechanism" happens in the agent's model $\mathcal{M}$ and not in the underlying system $\overline{\mathcal{M}}$, because $\overline{Y}$ does not have variables that could be the domain/codomain of a mechanism[2].

Let us formalize this kind of invariance:

**Definition 3.3** (Invariance of Determination). *Let $a, b$ be actions, $\bar{a}_J : Y_I \to Y_J$ a mapping, and assume that the determination relation $\text{outcome}_J^a = \bar{a}_J \text{outcome}_I^a$ holds. If $\text{outcome}_J^{ba} = \bar{a}_J \text{outcome}_I^{ba}$ also holds, we say that $b$ leaves the determination via $\bar{a}_J$ invariant.*

If in some context $X_c$ a certain determination via $\bar{s}_j : Y_{\text{Pa}_j^s} \to Y_j$ holds and is invariant to many later actions, this would be useful to know about and we could then call $\bar{s}_j$ a mechanism active in this context. The mechanism, along with the set of actions that leave it invariant, is thus a piece of knowledge about $X_c$ that can be used to reason about actions or other changes in this context. Relative to a set of mechanisms, an action can be viewed as a surgical intervention if it invalidates exactly one mechanism, and if in the new context a new mechanism is installed (meaning that the target variable is determined via this mechanism, and this relation is invariant).

## 3.3 STRUCTURAL CAUSAL MODELS

In our general setup, the variables, interventions, outcomes, and predictors/mechanisms could stand in all sorts of relations to each other, and we have sketched some desirable relations such as invariant determination and surgicality. Next we show how to encode an SCM in our framework, yielding a model with particularly simple relations. It should be noted though that for a general system it is not guaranteed that one can usefully model it in this way. Hence learning an SCM for a given set of variables is only part of the problem; finding the variables, figuring out which outcomes are

---

[2]One can perhaps find a natural isomorphism to a model with variables, but probably not canonically. In other words, as in physics, one should usually not view the coordinates used to describe a system as intrinsic.

possible and impossible, learning intervention skills, etc., is likely to be at least as important for AI.

Let $(U, V, F)$ be an SCM which we wish to encode in our framework. One defines $X = M \times U$, where $U = \prod_i U_i$ are exogenous variables and $M = \prod_i M_i$ is a space of mechanisms for each endogenous variable $V_i$. Each variable can be determined by its default (initialized / unintervened) mechanism $f_i$ or intervened on to equal a fixed value in $V_i$, so $M_i = V_i \cup \{f_i\}$. The space of outcomes is $Y = U \times V$, and the process is defined as $\text{proc}_Y(u, m) = (u, V_m(u))$, where $V_m(u)$ is the *potential response* defined by the SCM for intervention condition $m$ and exogenous $u$ (i.e. the solution to the equations indicated by $m$). One can define an initialization intervention $\text{do}_X(s)$ that maps everything to $m_0 = (f_1, \ldots, f_n)$. For each value of each $V_i$, one can define an effective intervention $v_i$ that sets the corresponding $M_i$ to that value, while leaving $U$ and other $M_j$ unchanged.

It is clear that interventions on different variables commute ($ab = ba$), interventions on the same variable overwrite ($ab = a$), and that $U$ is invariant. Furthermore, since the potential response is defined as the solution to a set of structural equations, the outcome $\text{proc}_Y(m, u)$ satisfies all the equations corresponding to $m$, as required in the original SCM $(U, V, F)$. Since $\text{do}_X(m_j)$ sets $M_j$ to a particular mechanism $g_j = f_j$ or $g_j = v_j$, it follows that the determination relation $\text{outcome}_j^{m_j} = g_j \text{outcome}_{\text{Pa}_j \times U_j}^{m_j}$ holds. It is easy to see that this determination relation is invariant.

If one has designed an SCM by hand, it is probably not useful to encode it in this way. However, when the causal variables, interventions, and mechanisms are to be learned from interactive experience, or when a more general kind of model of interventions is required (as in the general domino domain, where a single DAG is not enough), a setup like ours may be more appropriate. Furthermore, because our framework is based on maps, one automatically obtains a notion of natural transformation between models which can be used to define veridicality and model abstraction. Finally, this encoding shows that our models are easily general enough to describe any process described by SCMs, while allowing one to reason about the order of actions and more.

## 4   DISCUSSION & RELATED WORK

Although our work touches on a lot of topics, it was initiated to better understand the challenge of causal representation learning Schölkopf et al. [2021] and skill learning Eysenbach et al. [2018], Sharma et al. [2020], and their relation Bengio et al. [2017], Weichwald et al. [2022]. Causal representation learning has recently received a lot of attention Locatello et al. [2018], Locatello et al. [2020], Brehmer et al. [2022], Lippe et al. [2022], Mitrovic et al. [2020], Wang and Jordan [2021], Ke et al. [2020], Thomas et al. [2017]. Most works in this area focus on learning only the causal

representation, using various assumptions on the data generating process, sometimes including interventional data. As such, these works do not consider intervention policies.

Earlier theoretical work has identified and grappled with the problem of variable choice, but there is no complete theory yet [Eberhardt, 2016, Spirtes, Casini et al., 2021, Woodward, 2016]. As discussed in our paper, the notion of natural transformation coupled with a definition of intervention as mapping can be used to say what a permissible choice of variables is. A related issue is that of ambiguous manipulations (e.g. setting Total Cholesterol, whose outcome depends on the level of HDL and LDL cholesterol), and has been studied in Spirtes and Scheines [2004]. Defining an intervention as a mapping on a micro state-space completely eliminates ambiguity, although it is impractical for most scientific applications of causal modelling. Nevertheless our framework should be helpful in understanding the issue. The relation between causality and invariance was studied by Woodward [1997], Cartwright [2003], Peters et al. [2015], Arjovsky et al. [2019].

Although both atomic and soft interventions (replacing a mechanism) have been considered in the literature [Correa and Bareinboim, 2020], it was not known until now when one can describe a process at the microscopic ($\overline{X}$) level of description as an atomic or soft intervention in a causal model – a question of fundamental importance to causal representation & intervention skill learning. There is interesting work showing how causal models can emerge from systems of differential equations Mooij and Heskes [2013], Bongers et al. [2018], Blom and Mooij [2021], Rubenstein et al. [2016]. These works differ from ours in that we aim to describe interventions themselves as composable processes, and that our framework is more basic, relying on bare sets and maps instead of differential equations.

## 5   CONCLUSION

We have presented a natural theory of causation and intervention, based on the idea that an intervention must be a physically possible transformation of the state space of a system, for instance produced by an agent running a policy. We answer the question what it should mean for such a transformation to count as a surgical intervention setting an invariant mechanism for a variable. Our theory reconstructs the theory of SCMs, but grounds it in actual behaviours and generalizes it (for in our framework one can easily describe actions that are not surgical interventions, drastically change the graph, and express more knowledge about actions such as non-commutativity). Conceptually, the notion of intervention is clarified by giving it a concrete interpretation as a (physical) process, and mechanism as an invariant predictor. From an AI perspective, our work provides the beginnings of a theoretical foundation for causal representation & intervention skill learning.

## Acknowledgements

I would like to thank Pim de Haan, Johann Brehmer, Phillip Lippe, Sara Magliacane, Yuki Asano and Stratis Gavves for many interesting and inspiring discussions and Pim, Johann and Phillip for proofreading.

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
