# OpenReview forum: "Towards a Grounded Theory of Causation for Embodied AI"
_auai.org/UAI/2022/Workshop/CRL — CRL@UAI 2022 Poster_

### Official Review · Reviewer_awVi · 2022-06-26
**Review for Paper#36**

**Rating:** 4
**Confidence:** 3

**Review:**

This paper proposes a novel mathematical framework to represent "surgical interventions" which could lead to changes in the underlying system dynamics. The authors then introduce definitions of effects and invariances based on the proposed "surgical intervention" model. However, no inference algorithm is developed for this framework.

Overall, I found that the proposed method is not well motivated. The authors should include additional discussion about the placement of the proposed framework in the existing literature. For instance, there exist concepts like stochastic interventions (Correa & Bareinboin 2020) and dynamic treatment regimes (Murphy 2001) which seem to be sufficient in modeling "skills" in RL. How do these concepts compare with the proposed "surgical interventions"? I appreciate the running example about dominoes. However, its discussion could be improved. Personally, it appears that concepts like surgical interventions could be represented using existing methods. The motivation for the proposed framework is a bit unclear.

---

### Official Review · Reviewer_2tCW · 2022-07-03
**Review of "Towards a Grounded Theory of Causation for Embodied AI"**

**Rating:** 7
**Confidence:** 4

**Review:**

The authors introduce a framework for defining surgical interventions from a “grounded” perspective, where the interventions are grounded by actions in an action model. The grounding actions are required to satisfy properties which guarantee that the interventions behave modularly, expressed in terms of a vocabulary fof “determination relations” and “invariance” introduced in the paper. They further show how to construct an action model which grounds a given structural causal model.

To the best of my knowledge, this theory of grounding is original, and I expect it to be a useful and significant formalism for causal representation learning: it provides clear criteria for what would constitute a causal representation that could be learned from an action model. The paper is also clearly written/organized, and properly summarizes the related work of which I am aware. They also reference literature on action models/intervention skill learning, of which I am less familiar, but the coverage of related work in these areas also appears thorough.

In conclusion,

**Pros:**
- The paper provides an original and significant framework for grounding structural causal models in terms of action models.
- The paper is clearly written and adequately covers related work.
- The running example throughout the paper is very helpful for understanding the somewhat involved mathematical notation.
**Cons:**
- Nothing glaring, some minor suggestions below.
**Minor suggestions:**
- The authors mostly use \sqcap (or some close symbol) for Cartesian product, and \prod in one place. I think \times is more standard, and \prod is okay, but \sqcap looks odd.
- The authors (top right column of page 4) that “disentanglement can only occur if some parts of the representation space are impossible”. The “only” does not seem correct or to follow from anything written previously.
- In the left column of page 2, the authors refer to the figure “as shown on the right”, but the placement is below. The figure should also be captioned and referred to as Fig. 1.

---

### Meta-Review · Program_Chairs · 2022-07-06

**Recommendation:** Accept (Poster)
**Confidence:** 3

**Metareview:**

There appears to be some disagreement among the reviewers. While one thinks that "the paper provides an original and significant framework for grounding structural causal models in terms of action models", the other one complained about insufficient motivation and placement of the present contribution in the context of related work. Nevertheless, we believe that this framework grounding the theory of SCMs in actual behaviours may spark fruitful discussions in the workshop.

---

### Decision · Program_Chairs · 2022-07-06

Accept (Poster)